# The Relationship between Circulating Acetate and Human Insulin Resistance before and after Weight Loss in the DiOGenes Study

**DOI:** 10.3390/nu12020339

**Published:** 2020-01-28

**Authors:** Manuel A. González Hernández, Emanuel E. Canfora, Kenneth Pasmans, A. Astrup, W. H. M. Saris, Ellen E. Blaak

**Affiliations:** 1Department of Human Biology, NUTRIM School of Nutrition and Translational Research in Metabolism, Maastricht University Medical Centre+, Universiteitssingel 50, P.O. Box 616, 6229 ER Maastricht, The Netherlands; m.gonzalezhernandez@maastrichtuniversity.nl (M.A.G.H.); emanuel.canfora@maastrichtuniversity.nl (E.E.C.); k.pasmans@maastrichtuniversity.nl (K.P.); w.saris@maastrichtuniversity.nl (W.H.M.S.); 2Department of Human Nutrition, University of Copenhagen, 1171 Copenhagen, Denmark; ast@nexs.ku.dk

**Keywords:** short chain fatty acids, obesity, insulin resistance, weight loss

## Abstract

Microbially-produced acetate has been reported to beneficially affect metabolic health through effects on satiety, energy expenditure, insulin sensitivity, and substrate utilization. Here, we investigate the association between sex-specific concentrations of acetate and insulin sensitivity/resistance indices (Homeostatic Model Assessment of Insulin Resistance (HOMA-IR), circulating insulin and Matsuda Index) in the Diet, Obesity and Genes (DiOGenes) Dietary study at baseline and after a low-calorie diet (LCD, 800 kcal/d). In this analysis, 692 subjects (Body Mass Index >27 kg/m^2^) were included, who underwent an LCD for 8 weeks. Linear mixed models were performed, which were adjusted for mean acetate concentration, center (random factor), age, weight loss, and fat-free mass (FFM). At baseline, no associations between plasma acetate and insulin sensitivity/resistance indices were found. We found a slight positive association between changes in acetate and changes in HOMA-IR (stdβ 0.130, *p* = 0.033) in women, but not in men (stdβ −0.072, *p* = 0.310) independently of age, weight loss and FFM. We were not able to confirm previously reported associations between acetate and insulin sensitivity in this large European cohort. The mechanisms behind the sex-specific relationship between LCD-induced changes in acetate and insulin sensitivity require further study.

## 1. Introduction

Microbially-derived acetate has been shown to play an important role in substrate and energy metabolism [1]. In rodents fed high-fat diets, beneficial effects on insulin sensitivity [2] and increased white adipose tissue browning have been reported [3]. However, rodent data are not entirely consistent since intragastric acetate infusions in rats have resulted in hyperphagia and energy retention [4].

In humans, rectal acetate infusions have been shown to increase plasma peptide YY (PYY) and glucagon-like peptide-1 in hyperinsulinaemic overweight women [5]. Furthermore, acute distal colonic acetate infusions and short chain fatty acids (SCFA) mixtures high in acetate decreased whole-body lipolysis, increased fasting fat oxidation, resting energy expenditure, and PYY secretion in overweight men [6,7]. Additionally, circulating acetate was positively associated with increased fat oxidation and energy expenditure [7]. Recently, a study reported that acetate produced in the distal colon contributes significantly to systemic acetate concentrations in men [8]. In line with this, cross-sectional analyses in patients with obesity have shown positive associations between high circulating acetate and insulin sensitivity indices [9,10]. In women, plasma acetate correlated negatively with fasting and postprandial insulin concentrations [9]. In morbidly obese subjects, acetate correlated negatively with Homeostatic Model Assessment of Insulin Resistance (HOMA-IR) [10]. Notably, a resistant starch supplementation (30 g/d for 4 weeks) in healthy subjects increased acetate uptake in muscle and adipose tissue and improved insulin sensitivity (as measured by hyperinsulinemic-euglycemic clamp) [11].

Here, we assess the association of (changes in) circulating acetate and insulin sensitivity/resistance indices (HOMA-IR, circulating insulin and Matsuda Index) at baseline and after an eight-week low calorie diet (LCD) in male and female overweight/obese subjects of the European Diet, Obesity and Genes (DiOGenes) Dietary study. Due to the fact that microbial composition and functionality, as well as their impact on insulin sensitivity, may differ in overweight males and females, we stratified our analysis for sex [12].

## 2. Materials and Methods

### 2.1. Study Design

The DiOGenes project is a multicenter, randomized, controlled dietary intervention study in 8 European countries. In total, 938 overweight or obese, nondiabetic adults, free of cardiovascular disease (age 18–65 years, body mass index (BMI) 27–45 kg/m^2^ and fasting blood glucose concentrations <6.1 mmol/L) were recruited. Subjects using prescription medication or suffering from diseases or conditions that might influence the outcome of the study were excluded. Of special interest were diseases that influenced body weight regulation (e.g., malabsorption, untreated hypo/hyperthyroidism, eating disorders, systemic use of steroids) and obesity-related cardiovascular risk factors (heart disease, systolic and diastolic blood pressures ≥160/100 mmHg, blood glucose >6.1 mmolL^−1^, blood cholesterol >7 mmolL^−1^, blood triglycerides >3 mmolL^−1^). More details on recruitment, inclusion, and exclusion criteria and study design are described elsewhere [13].

The Medical Ethical Committees of the respective countries approved the study protocol, which followed the Helsinki II regulations. The study was registered with ClinicalTrials.gov, number NCT00390637. Succinctly, the study consisted of two phases. In this report, we focus on the baseline characterization of the overweight/obese individuals as well as the weight-loss phase in which individuals with BMI >27 kg/m^2^ were assigned to lose weight following an LCD for 8 weeks. The LCD provided 3.3 MJ/d (800 kcal/d) using Modifast products (Nutrition et Santé, France, area code 31451), with a macronutrient composition of 15–20% of total energy from fat, 35–40% from protein, and 45–50% from carbohydrate. In addition, subjects could also eat up to 400 g of vegetables, providing a total of 3.3 to 4.2 MJ/d (800 to 1000 kcal/d).

### 2.2. Study Population

We included 692 volunteers (441 women, 251 men) from whom plasma metabolomics (including acetate) were available before and after the LCD. Volunteers with BMI >27 kg/m^2^ were recruited in eight different research centers across Europe: Maastricht (the Netherlands), Copenhagen (Denmark), Cambridge (United Kingdom), Heraklion (Greece), Potsdam (Germany), Pamplona (Spain), Sofia (Bulgaria), and Prague (the Czech Republic). All subjects provided written informed consent before enrollment into the study.

### 2.3. Anthropometry and Blood Sampling

Anthropometry and blood samples were collected after fasting (10 hours) at baseline (clinical investigation day 1, CID1) and after weight loss at the end of the LCD period (CID2). In this report, we included the following parameters: anthropometry (body weight, BMI, and fat-free mass (FFM)). FFM was assessed by multifrequency bioimpedance (QuadScan 4000, Bodystat, Douglas, United Kingdom) [13]. Additionally, glucose, free fatty acids (FFA) (automatic spectrophotometric enzymatic techniques) and insulin were measured from fasting samples [13]. Insulin concentrations were measured by a colorimetric assay (Ortho-Clinical Diagnostics, Johnson & Johnson, Birkerød, Denmark) [14]. Homeostatic Model Assessment of Insulin Resistance (HOMA-IR) and insulin sensitivity (Matsuda index) were calculated. HOMA-IR was calculated as follows: HOMA-IR = (glucose (mmol/L) * insulin (µU/mL)/22.5), using fasting values [15]. Matsuda index was calculated as follows: (10,000/square root of (fasting glucose x fasting insulin) x (mean glucose x mean insulin during Oral glucose tolerance test (OGTT))) [16]. BMI was calculated by dividing the mass in kg by squared height. FFM was obtained from bioimpedance analysis. Acetate, acetoacetate, and 3-OH-butyrate were quantified in serum from nuclear magnetic resonance (NMR) spectra, as reported previously in detail [17]. Physical activity was measured before and after LCD by means of the Baecke questionnaire [13].

### 2.4. Statistical Analysis

The normality of data was assessed with the Kolmogorov–Smirnov procedure and histogram and variables that were not normally distributed were Ln-transformed. Repeated measures ANOVA with Bonferroni correction was conducted to investigate sex, time, and sex–time effects as a result of the LCD phase. A linear mixed model (LMM) was used to determine the relationship between (changes in) acetate and (changes in) insulin sensitivity/resistance parameters at baseline and after the LCD. In the first model, besides (changes in) acetate and (changes in) fasting insulin, HOMA-IR or Matsuda index, mean acetate was added. Subsequently, age (Model 2), (change in) weight (Model 3), and (change in) FFM (Model 4) were added as covariates. In all models, the center was included as a random factor.

## 3. Results

### 3.1. Cross-Sectional Analysis at Baseline

Acetate showed significantly higher levels in men as compared to women at baseline (1.35 ± 1.39 vs. 1.13 ± 0.97 mmol/L, respectively, see Table 1). There were no differences in BMI between males and females. Furthermore, fasting insulin, HOMA-IR, fasting glucose, body weight and FFM were higher in males, whilst the Matsuda index, FFA, and ketone bodies were lower (see Table 1). There were no relationships between acetate and fasting insulin, HOMA-IR, and Matsuda Index in males as well as in females.

### 3.2. Weight Loss by LCD

In response to the LCD, acetate levels increased in both men and women (1.35 ± 1.39 to 1.45 ± 1.16 and 1.13 ± 0.97 to 1.25 ± 1.03 mmol/L, respectively, Table 1), but did not differ between sexes (*p* = 0.852). Weight reduction increased insulin sensitivity, as assessed by Matsuda (*p* = 0.002) and decreased HOMA-IR (*p* = 0.004) and fasting insulin (*p* = 0.001), with more pronounced improvements in males as compared to females. Additionally, FFA slightly increased as a result of weight loss with a more pronounced increase in males as compared to females (Table 1). Ketone bodies (acetoacetate and 3-OH-butyrate) showed an increase while acetate/3-OH-butyrate and acetate/acetoacetate ratios decreased after LCD; however, sex–time interactions were not significant. Physical activity (sports and work indexes) did not change after LCD. Leisure activity differed (sex and time); however, sex-time interaction was not significant.

We observed a slight positive relationship between LCD-induced changes in acetate concentrations and changes in HOMA-IR (stdß 0.130, *p* = 0.033, Table 2) and with changes in fasting insulin concentrations (stdß 0.119, *p* = 0.051) in women, but not in men (stdß −0.072, *p* = 0.310 and stdß −0.066, *p* = 0.359, respectively; data not shown). Subsequently, the relationship between (changes in) acetate and HOMA-IR did not change after adjustment for age and changes in body weight as well as changes in FFM (Table 2). A similar result was seen for (changes in) fasting insulin (Table 3). The relationship between changes in acetate and Matsuda index did not reach statistical significance (data not shown).

## 4. Discussion

In the present study, we could not confirm previously reported positive associations between acetate and insulin sensitivity and metabolic health parameters in a cross-sectional analysis at baseline. LCD-induced weight loss increased insulin sensitivity more in men than in women. Surprisingly, we found a slight positive association between LCD-induced changes in acetate and changes in HOMA-IR and fasting insulin concentrations in overweight/obese women, but not in men, which was independent of age and changes in body composition.

As previously described, cross-sectional studies showed negative associations between acetate and fasting insulin in obese women [9]. Similarly, in morbidly obese subjects, a negative association with HOMA-IR and a positive association with insulin sensitivity were observed [10]. Furthermore, acute distal colonic acetate infusions increased fat oxidation and improved metabolic profile in overweight/obese men [6,7]. Surprisingly, we observed a slight positive relationship between the LCD-induced change in circulating acetate and change in HOMA-IR and fasting insulin in females, but not in males. In line with these results on differential associations between males and females, a previous DiOGenes analysis by Stroeve et al. [17], suggested a sex-specific modulatory role of baseline acetate in energy metabolism, since they reported a negative association with LCD-induced BMI change in morbidly obese females (stdß -0.23), but not in males. Furthermore, a previous analysis in the DiOGenes study showed that despite the fact that hepatic insulin resistance was lower in females as compared to males, there was a more pronounced worsening of blood lipid profile with the progression of hepatic insulin resistance in females, but not in males [18]. In interpreting our results, it is important to note that acetate kinetics (acetate absorption in the colon, hepatic uptake and acetate clearance), and turnover (production/utilization) are intricate processes that are influenced by the interplay of gut microbial fermentation, lipogenesis, lipid oxidation and ketogenesis [19]. With respect to the insulinemic profile, hyperinsulinemia may affect acetate metabolism, possibly via an alteration in endogenous and exogenous acetate metabolism [20]. In addition, a caloric restriction may augment hepatic ketogenesis via an overproduction of acetyl-coA coming from increased lipid utilization [21] and increased ketogenesis may also elevate acetate production [22].

The exact mechanisms behind the sex-specific relationship of acetate and HOMA-IR may relate to sex hormones affecting processes such as microbial metabolism, hepatic lipid metabolism, and ketogenesis. Notably, in our study, most women were in the premenopausal state; however, data on the hormone status was not available. Ketone bodies increased after the LCD with no differences between males and females. In addition, acetate/3-OH-butyrate and acetate/acetoacetate ratios decreased after LCD, which may suggest that the contribution of acetate from sources other than hepatic ketogenesis has probably not increased after LCD [23]. Thus, the discrepancy of our findings on the relationship between acetate and insulin resistance as well as the sexual dimorphism in this relationship remains to be elucidated, taking into account the complex acetate kinetics. Finally, other SCFA (propionate and butyrate) may be of importance for insulin sensitivity, as we and other publications have reported [24,25,26,27,28]. Unfortunately, these metabolites were not included in the NMR measurements and we focused on acetate, the most abundant SCFA in the colonic lumen and systemic circulation.

The major strength of our study is the availability of HOMA-IR and a 5-point oral glucose tolerance test (including insulin concentrations for calculation of the Matsuda index) in a large and well-characterized cohort, which made additional adjustment for confounding factors feasible. A limitation of our study is the lack of information on acetate kinetics and colonic acetate that may contribute to circulating acetate concentrations. Of note, in the current study, we measured plasma acetate concentrations using NMR methodology, of which validation has been previously described [17,29,30]. Although the absolute concentrations of acetate may differ as compared to previous studies measured with LC-MS or GC-MS [6,7,8,9,25,31,32], high correlations have been reported between metabolites measured with both LC-MS and NMR [33].

In conclusion, the results of this large pan-European study (692 volunteers) showed no relationship between acetate and markers of insulin sensitivity. Furthermore, a small positive association between LCD-induced changes in acetate and HOMA-IR and fasting insulin (explaining 1–2% of the variance in HOMA-IR or fasting insulin) was shown in females and not in males. Future studies should aim to elucidate the underlying mechanisms and physiological significance.

## Figures and Tables

**Table 1 nutrients-12-00339-t001:** Characteristics of participants before and after low calorie diet (LCD) intervention.

	Men		Women		Effect-Sex	Time	Time × Sex
	Baseline	After LCD	Baseline	After LCD			
*n*	175	175	303	303			
Age (yr)	42 ± 6	42 ± 6	41 ± 6	41 ± 6	0		1
Acetate (mmol/L)	1.36 ± 1.54	1.41 ± 1.25	1.12 ± 1.03	1.18 ± 1.04	0.000	0.016	0.672
HOMA index	3.86 ± 2.35	2.18 ± 2.23	2.93 ± 3.0	2.00 ± 2.53	0.000	0.000	0.004
Matsuda index	4.10 ± 2.50	7.16 ± 3.8	5.67 ± 3.50	7.55 ± 3.70	0.000	0.000	0.002
Insulin (µU/mL)	13.8 ± 7.6	8.1 ± 5.3	11.14 ± 11.14	8.18 ± 9.32	0.000	0.000	0.001
Glucose (mmol/L)	5.3 ± 0.6	5.0 ± 0.5	5.0 ± 0.6	4.8 ± 0.3	0.000	0.000	0.457
FFA (micromol/L)	528 ± 190	634 ± 204	675 ± 263	746 ± 213	0.000	0.000	0.176
Acetoacetate (mmol/L)	0.06 ± 0.04	0.23 ± 0.30	0.09 ± 0.07	0.25 ± 0.30	0.000	0.000	0.376
3-OH-butyrate (mmol/L)	0.35 ± 0.30	1.52 ± 2.00	0.56 ± 0.52	1.69 ± 1.94	0.000	0.000	0.133
Acetate/3-OH-butyrate ratio	5.80 ± 8	2.08 ± 3	3.50 ± 4.90	1.40 ± 2.04	0.000	0.000	0.415
Acetate/Acetoacetate ratio	26 ± 29	12 ± 16	18 ± 21	9 ± 10	0.000	0.000	0.787
Body weight (kg)	111.8 ± 17.5	99.0 ± 15.9	96.8 ± 16.5	86.6 ± 15	0.000	0.000	0.702
BMI (kg/m^2^)	35 ± 4.8	31 ± 4.4	35.2 ± 5.1	31.4 ± 4.6	0.259	0.000	0.657
Fat-free mass (kg)	73.4 ± 9.9	69.8 ± 8.7	53.4 ± 8.1	51 ± 8	0.000	0.000	0.990
Baecke Questionnaire scores							
Leisure index	2.6 ± 0.7	2.8 ± 0.6	2.70 ± 0.7	3.07 ± 0.6	0.000	0.001	0.569
Sports index	2.6 ± 0.4	2.6 ± 0.4	2.6 ± 0.5	2.7 ± 0.4	0.620	0.675	0.310
Work index	2.7 ± 0.4	2.8 ± 0.4	2.7 ± 0.4	2.7 ± 0.3	0.087	0.670	0.980

Repeated measures ANOVA. First *p*-value reported corresponds to sex effect. Second *p*-value corresponds to the time effect and third *p*-value corresponds to the sex–time effect. Data expressed as mean and standard deviation. Abbreviations: low-calorie diet (LCD), Homeostatic Model Assessment of Insulin Resistance (HOMA-IR), BMI (Body Mass Index) free fatty acids (FFA).

**Table 2 nutrients-12-00339-t002:** Determinants of changes in HOMA-IR in participants during weight loss.

Model	Parameter	Females		Males	
		Std ß, Confidence Interval	*p*-Value	Std ß, Confidence Interval	*p*-Value
1	Δ-Acetate Mean acetate	0.111 (0.013 to 0.209)	0.027	−0.098 (−0.224 to 0.027)	0.123
2	Model 1 + Age	0.125 (0.027 to 0.223)	0.013	−0.101 (−0.227 to 0.025)	0.115
3	Model 2 + Δ-Weight	0.120 (0.022 to 0.218)	0.017	−0.069 (−0.189 to 0.051)	0.259
4	Model 3 + Δ-Fat-free mass	0.130 (0.010 to 0.249)	0.033	−0.072 (−0.211 to 0.068)	0.310

Linear mixed model was adjusted for age, weight, and fat-free mass. All models were adjusted for center as a random factor (coefficients not shown). Acetate concentration (independent factor) and HOMA-IR as dependent factors. Statistically significant *p*-values are in bold. Valid cases *n* = 302 females, 175 males.

**Table 3 nutrients-12-00339-t003:** Determinants of changes in fasting insulin in participants during weight loss.

Model	Parameter	Females		Males	
		Std ß, Confidence Interval	*p*-Value	Std ß, Confidence Interval	*p*-Value
1	Δ-Acetate Mean acetate	0.132 (0.034 to 0.231)	0.009	−0.085 (−0.212 to 0.042)	0.190
2	Model 1 + Age	0.140 (0.042 to 0.238)	0.005	−0.089 (−0.217 to 0.038)	0.168
3	Model 2 + Δ-Weight	0.132 (0.035 to 0.229)	0.008	−0.064 (−0.188 to 0.059)	0.306
4	Model 3 + Δ-Fat-free mass	0.119 (−0.001 to 0.238)	0.051	−0.066 (−0.207 to 0.075)	0.359

Linear mixed model was adjusted for age, weight, and fat-free mass. All models were adjusted for center as a random factor (coefficients not shown). Acetate concentration (independent factor) and HOMA-IR as dependent factors. Statistically significant *p* values are in bold. Valid cases *n* = 295 females, 175 males.

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
