# Peer review of "The Relationship between Circulating Acetate and Human Insulin Resistance before and after Weight Loss in the DiOGenes Study"

_nutrients, 2020, doi:10.3390/nu12020339_

Round 1
Reviewer 1 Report
Thanks for taking the time to answer my questions. I'm happy with your responses and I do have further comments. Congratulations.
Reviewer 2 Report
The authors satisfied the reviewer's comments.
This manuscript is a resubmission of an earlier submission. The following is a list of the peer review reports and author responses from that submission.
Round 1
Reviewer 1 Report
The purpose of the study was to determine the relationship between acetate levels and insulin resistance in human males and females. The study pulled data from the DiOGenes program, resulting in a reasonably sized cohort. The manuscript was well written and the data were clearly presented. In summary, there lacked a relationship between acetate levels insulin responsiveness. Findings were also not consistent with other trends reported in the literature. This includes the relationship between LCD and acetate levels. There were several findings that appeared to be sexually dimorphic, yet there is no discussion about that. The study would benefit by addressing such differences in the discussion by speculating, at a minimum, about the mechanisms contributing to the sexually dimorphic responses.
Reviewer 2 Report
Overall, an interesting study with descriptive information, which does not help to explain anything mechanistically. Part of the discrepancies from studies may be due to differences in energy balance and exercise (for example). A low energy balance is typically associated with peripheral insulin resistance, increased lipolysis, increased FFA (which also contribute the peripheral insulin resistance), low glucose due to the low energy balance and low total intake of carbohydrates, etc. This promotes fat oxidation while preserving glucose for glucose-dependent tissues. Under these circumstances, neither HOMA-RI nor resting insulin are useful markers of peripheral insulin sensitivity. This complicates the interpretation of this study. It will also be good to know what happened with propionate and butyrate, which were likely measured in the context of this study.
Major
Information regarding physical activity should be included and taken into account as a potential confounding variable. Please, also explain in which conditions were blood samples taken. Had the subjects any co-morbidities (for example one would expect 15-25% of subjects to have high blood pressure, did you include subjects with type 2 diabetes)? What about other medical treatments (statins, antihypertensive drugs, etc.). How many women were pre/post-menopausal?
Do you have any information regarding metabolism t rest: RMR, RQ and substrate oxidation by indirect calorimetry?
On the other hand, you can look at the ratio acetate/Beta-hydroxybutyrate. This ratio es expected to increase when the acetate increases in the plasma due to sources of acetate different from ketogenesis.
Minor
L24: “stdß”
L83 The kits used to measure insulin should be reported
L88 The bioimpedance device and equations used to estimate FFM should be reported. BIA is somewhat inaccurate and depends on population-specific equations to estimate FFM: do these equations work likely well in the different European countries?
L89: You used NMR spectra for ketone bodies: can you provide additional information on the validity of the quantitative data reported?
Table 2. N is different than in Table 1: why?